# Mesh-TensorFlow:
# Deep Learning for Supercomputers

**Noam Shazeer, Youlong Cheng, Niki Parmar,**
**Dustin Tran, Ashish Vaswani, Penporn Koanantakool, Peter Hawkins, HyoukJoong Lee**
**Mingsheng Hong, Cliff Young, Ryan Sepassi, Blake Hechtman**
Google Brain
{noam, ylc, nikip, trandustin, avaswani, penporn, phawkins,
hyouklee, hongm, cliffy, rsepassi, blakehechtman}@google.com

## Abstract

Batch-splitting (data-parallelism) is the dominant distributed Deep Neural Network (DNN) training strategy, due to its universal applicability and its amenability to Single-Program-Multiple-Data (SPMD) programming. However, batch-splitting suffers from problems including the inability to train very large models (due to memory constraints), high latency, and inefficiency at small batch sizes. All of these can be solved by more general distribution strategies (model-parallelism). Unfortunately, efficient model-parallel algorithms tend to be complicated to discover, describe, and to implement, particularly on large clusters. We introduce Mesh-TensorFlow, a language for specifying a general class of distributed tensor computations. Where data-parallelism can be viewed as splitting tensors and operations along the "batch" dimension, in Mesh-TensorFlow, the user can specify any tensor-dimensions to be split across any dimensions of a multi-dimensional mesh of processors. A Mesh-TensorFlow graph compiles into a SPMD program consisting of parallel operations coupled with collective communication primitives such as Allreduce. We use Mesh-TensorFlow to implement an efficient data-parallel, model-parallel version of the Transformer [16] sequence-to-sequence model. Using TPU meshes of up to 512 cores, we train Transformer models with up to 5 billion parameters, surpassing state of the art results on WMT'14 English-to-French translation task and the one-billion-word language modeling benchmark. Mesh-Tensorflow is available at https://github.com/tensorflow/mesh .

## 1   Introduction

Batch-splitting (data-parallelism) is the dominant distributed Deep Neural Network (DNN) training strategy, due to its universal applicability and its amenability to Single-Program-Multiple-Data (SPMD) programming. However, batch-splitting suffers from several major problems when training very large models. The memory required to store parameters and/or activations and the time necessary to synchronize parameters can make purely-data-parallel algorithms impossible or inefficient. Different distribution strategies (model-parallelism [7]) can solve these issues, but specifying these strategies can be complicated, and the current MIMD implementations generate very large programs which can be difficult to compile and to optimize.

We solve this problem by introducing Mesh-TensorFlow, a language for specifying a general class of distributed tensor computations. Where data-parallelism can be viewed as splitting tensors and operations along the "batch" dimension, in Mesh-TensorFlow, the user can specify any tensor-dimensions to be split across any dimensions of a multi-dimensional mesh of processors. A Mesh-TensorFlow graph compiles into a SPMD program consisting of parallel operations coupled with

collective communication primitives such as Allreduce. We use Mesh-TensorFlow to implement an efficient data-parallel, model-parallel version of the Transformer [16] sequence-to-sequence model. Using TPU meshes of up to 512 cores, we train Transformer models with up to 5 billion parameters, surpassing state-of-the-art results on WMT'14 English-to-French translation task and the one-billion-word Language modeling benchmark.

## 2 Hardware Assumptions

While much work deals with heterogeneous and/or unreliable hardware, we focus on clusters of identical, reliable processors, each with a local memory. We define a **mesh** as an n-dimensional array of such processors. The mesh is only a naming abstraction and does not imply a physical network topology. As such, different meshes can be defined over the same set of physical processors. For example, a 512-core TPU cluster with a 16x16x2 toroidal network interconnect could be represented by a 3-dimensional mesh with shape [16, 16, 2], a two-dimensional mesh with shape [32, 16], a one-dimensional mesh with shape [512], etc. The physical network topology *does* affect performance; particularly important is the performance of MPI *Allreduce*, grouped by splitting the mesh by a subset of the dimensions, which can be very efficient [4] [5] if each such group is physically connected.

## 3 Inspiration: Single-Program-Multiple-Data (SPMD) Batch-Splitting

We first review a commonly-used variant of synchronous data-parallelism where each processor keeps an identical copy of all parameters (Algorithm 1). For each step, the batch of training examples is split into sub-batches, one for each processor. Each processor computes the forward and backward passes on its sub-batch, resulting in gradients on the model parameters. These gradients are then summed across all processors and the results broadcast to all processors (MPI-allreduce). Finally, each processor updates its own copy of the parameters.

---

**Algorithm 1** Synchronous data-parallelism with replicated parameters. Each processor maintains a complete copy of all weights $W^{(t)}$. The batch $b^{(t)}$ of training examples for timestep $t$ is partitioned among the set $P$ of processors: $b^{(t)} = \dot{\bigcup}_{p \in P} b_p^{(t)}$. Below is the computation performed on one processor $p \in P$.

---

1: Compute partial parameter gradients $\nabla Q(W^{(t)}, b_p^{(t)})$            ▷ Local computation
2: $\nabla Q(W^{(t)}, b^{(t)}) = \sum_{p' \in P} \nabla Q(W^{(t)}, b_{p'}^{(t)})$            ▷ *Allreduce*
3: $W^{(t+1)} = Update(W^{(t)}, \nabla Q(W^{(t)}, b^{(t)}))$            ▷ Local computation

---

This algorithm is typically implemented using Single-Program-Multiple-Data (SPMD) programming, with every processor running the same program of local operations and MPI-allreduce primitives.

One way to see this algorithm is that every tensor and every operation in the computation is either split across all processors (if it has a "batch" dimension), or fully replicated across all processors (if it does not have a "batch" dimension). Operations which reduce out the "batch" dimension require an additional MPI-allreduce to produce the correct result. We can describe this as splitting the computation across the "batch" dimension. Mesh-TensorFlow generalizes this idea to splitting computations across arbitrary dimensions.

## 4 Mesh-TensorFlow: Beyond Batch Splitting

Mesh-Tensorflow generalizes from the batch-splitting algorithm described in section 3 to allow for splitting across different Tensor dimensions. The similarities are as follows:

- Each tensor in the computation is represented by one (not-necessarily-distinct) slice of the tensor on each processor.
- Each operation in the computation is implemented as one operation on each processor. Most operations require no communication, with each processor producing its slice of the output from its slices of the inputs. Some operations additionally require collective communication primitives such as MPI-allreduce.

- Based on the above, the computation can be implemented as a SPMD program.

The new elements in Mesh-TensorFlow are as follows:

- Tensors have named dimensions. This allows for the idea of a logical dimension (like "batch") which will be split in the same way for different tensors and operations. It is illegal for a tensor to have two identically-named dimensions.

- Rather than an unstructured set of processors, Mesh-Tensorflow allows for an n-dimensional mesh of processors (section 2). The mesh also has named dimensions.

- A global "computation layout" is a partial map from tensor-dimension to mesh-dimension specifying which tensor-dimensions are split across which dimensions of the processor-mesh. For example, batch-splitting (data-parallelism) would be expressed by using a one-dimensional mesh with dimension `"all_processors"` and using the computation layout `[("batch", "all_processors")]`. This means that all tensors with a `"batch"` dimension are split along that dimension across all processors, while all other tensors are fully replicated.

## 5  Tensor Representations

A tensor is represented as one slice of the tensor per processor. The **layout** of a tensor is an injective partial map from the tensor's dimensions to dimensions of the mesh, and is computed as the restriction of the global computation layout to that tensor's dimensions. It is illegal for two dimensions of the same tensor to map to the same mesh dimension. If a tensor's layout is empty, it is fully replicated on each processor. For every (tensor-dimension, mesh-dimension) pair in the tensor's layout, the slice on a processor is restricted along that tensor-dimension to a stripe corresponding to that processor's coordinate along that mesh-dimension. The current implementation of Mesh-TensorFlow requires the size of the tensor-dimension to be evenly divisible by the size of the mesh-dimension.

## 6  Operation Implementation

Each operation is implemented by parallel computation on every processor, and sometimes collective communication. We describe the implementations of some important operations here:

**Component-wise Operations**   Mesh-TensorFlow supports component-wise operations where the shapes (and hence the layouts) of the input and output tensors are identical. These are trivially implemented by parallel operations on each processor to compute that processor's slice of the output from that processor's slice(s) of the input(s).

**Reduction (reduce_sum(), reduce_max(), etc.)**   Mesh-TensorFlow supports reductions where the output dimensions are a subset of the input dimensions. These can be implemented by local reductions of each slice, followed by MPI-allreduce across any mesh dimensions corresponding to reduced-out Tensor dimensions. The allreduce operation is necessary because the local reduction only sums across a subset of the split tensor-dimension. Bandwidth-efficient implementations of allreduce exist when the processors for each group are connected in any type of tree. [4] [5]

**Einstein Summation (matrix multiplication, etc.)**   Einstein-summation (einsum) notation (as defined in numpy, TensorFlow, etc.) is a way of expressing a class of operations including (batch) matrix multiplication, reductions and broadcasts, where the operation is defined by the names of the dimensions of the input and output tensors. Mesh-TensorFlow's use of named dimensions makes using einsum particularly convenient. Einsum can be defined as broadcasting all inputs to a shape consisting the union of all their dimensions, multiplying them component-wise, then reducing out all dimensions not in the specified output shape. Einsum is implemented by parallel einsum operations on each processor of that processor's input slices, followed by MPI-allreduce across any mesh dimensions corresponding to reduced-out Tensor dimensions.

### 6.1 Reshape

While reshape is simple in the non-distributed case, Mesh-TensorFlow reshape can require network communication, since the layout of the output tensor may differ from that of the input tensor. Even keeping the same dimension sizes, changing the dimension names (and hence the layout) can result in several different communication patterns: If a dimension is split in the input but not in the output, the implementation involves MPI-allgather communication across the corresponding mesh-dimension. If a dimension is split in the output but not in the input, the implementation involves no communication, just slicing on each processor. MPI-alltoall is used in the case where different dimensions in the input and the output are split across the same mesh dimension, as might be the case when switching between data-parallelism and model-parallelism for different layers of the same model, as in [15].

## 7 Mesh-TensorFlow syntax

The Mesh-TensorFlow language is nearly identical to TensorFlow [12], with the familiar notions of graphs, tensors, operations, variables, devices (called meshes), and automatic gradient computation. The principal difference is that in Mesh-TensorFlow, tensor-dimensions have a name as well as a size. The shape of each tensor is a statically-known tuple of such dimensions. Shapes are inferred automatically when possible, as they are in TensorFlow. Binary component-wise operations like addition employ implicit broadcasting in the case where the shape of one operand is a subset of the shape of the other.

The initial implementation of Mesh-TensorFlow is a Python library. The user builds a Mesh-TensorFlow graph in python, which the library "lowers" to generate part of a TensorFlow graph. As of the writing of this paper, implementations exist for generating SPMD TensorFlow code for TPUs, or MIMD code (using device placement) for multi-CPU/GPU configurations.

## 8 Example: Two Fully-Connected Layers

We consider a simple example of two fully-connected layers in the middle of a neural network. The input layer $x$ and the output layer $y$ each have $d_{io}$ units, and the hidden layer $h$ has $d_h$ units. The hidden layer also has a bias and $Relu$ activation.

$$y = Relu(xw + bias)v \tag{1}$$

This Mesh-TensorFlow code fragment runs these layers on a batch $x$ of $batch\_size = b$ inputs.

```
...
batch = mtf.Dimension("batch", b)
io = mtf.Dimension("io", d_io)
hidden = mtf.Dimension("hidden", d_h)
# x.shape == [batch, io]
w = mtf.get_variable("w", shape=[io, hidden])
bias = mtf.get_variable("bias", shape=[hidden])
v = mtf.get_variable("v", shape=[hidden, io])
h = mtf.relu(mtf.einsum(x, w, output_shape=[batch, hidden]) + bias)
y = mtf.einsum(h, v, output_shape=[batch, io])
...
```

The code above defines only the mathematical model. We now discuss several different computation layouts. Each will produce identical results, but will have different performance characteristics. We also provide illustrations of the layouts in the supplementary materials (Section S.1).

### 8.1 Data-Parallel Layout

To train the above model in data-parallel mode on a mesh of $n$ processors, we would define:

```
mesh_shape = [("all", n)]
computation_layout = [("batch", "all")]
```

When the Mesh-TensorFlow graph is compiled with this layout, the parameter tensors $w$, $v$, and $bias$ are replicated on all processors, but the activation matrices $x$, $h$, $y$, etc. are split across the batch dimension. For example, each processor keeps a slice of $x$ with shape $[\frac{b}{n}, d_{io}]$.

There is no inter-processor communication in the forward pass. However, the gradient computations for the parameters are `mtf.einsum` operations which reduce out the `batch` dimension, and hence produce *Allreduce* operations when they are compiled. The number of values allreduced per processor is equal to the number of parameters, approximately $2d_{io}d_h$.

## 8.2 Model-Parallel Layout

Rather than splitting the batch, we can split the units in the hidden layer:

```
mesh_shape = [("all", n)]
computation_layout = [("hidden", "all")]
```

When the Mesh-TensorFlow graph is compiled with this layout, the input and output layers $x$, and $y$ are replicated on all processors, but the hidden activations $h$ and the parameter tensors $w$, $v$ and $bias$ are all split across the `hidden` dimension. For example, each processor keeps a slice of $w$ with shape $[d_{io}, \frac{d_h}{n}]$ and a slice of $v$ with shape $[\frac{d_h}{n}, d_{io}]$.

When computing $y$, the split hidden dimension is reduced out. Consequently, the results of that computation get allreduced across all processors. A similar allreduce happens in computing the gradients on $x$. In all, the number of values allreduced per processor is $2bd_{io}$.

## 8.3 Data-Parallel, Model-Parallel Layouts

On a two-dimensional mesh of $r \times c$ processors, we can employ both data-parallelism and model-parallelism:

```
mesh_shape = [("rows", r), ("cols", c)]
computation_layout = [("batch", "rows"), ("hidden", "cols")]
```

In this layout, each row of processors handles a fraction of the batch, while each column of processors handles a fraction of the hidden units. Each processor keeps a slice of x with shape $[\frac{b}{r}, d_{io}]$, with processors in the same row having identical slices. The hidden activation tensor $h$ is tiled in two dimensions, with each processor keeping a slice with shape $[\frac{b}{r}, \frac{d_h}{c}]$.

This layout causes partitioned-allreduce operations in several places. For example, in computing $y$, we reduce out the `hidden` dimension, which is split over the `cols` dimension of the mesh, so the results of the operation need to be summed up by processor-column, as opposed to over the entire mesh. In all, the number of values allreduced per processor is $\frac{2bd_{io}}{r} + \frac{2d_{io}d_h}{c}$

If we have a three-dimensional mesh of processors, we can even split the computation in three dimensions:

```
mesh_shape = [("rows", r), ("cols", c), ("planes", p)]
computation_layout = [
  ("batch", "rows"), ("hidden", "cols"), ("io", "planes")]
```

In this case, every matrix in the computation is tiled across two mesh dimensions and replicated in the third, and every `einsum` requires an allreduce across one mesh dimension.

## 8.4 Inefficient Layouts

For a computation layout to be efficient, all expensive operations need to be split (as opposed to replicated) across all mesh dimensions. For example, the empty layout below produces correct results, but since it replicates all computation on every processor, it saves no time or memory. A general rule is that any expensive `einsum` operation should have one input dimension that is split across each batch dimension.

```
mesh_shape = [("all", n)]
computation_layout = []
```

## 8.5 Illegal Layouts

The computation layout below is illegal, because it causes the tensor $h$ to have two dimensions which are split across the same dimension of the mesh.

```
mesh_shape = [("all", n)]
computation_layout = [("batch", "all"), ("hidden", "all")]
```

## 8.6 Performance Comparison

| Layout | Comp. Time | Comm. Time | $\frac{communication}{computation}$ | Memory/Processor |
|---|---|---|---|---|
| [] | $bd_{io}d_h$ | 0 | 0 | $bd_{io} + bd_h + d_{io}d_h$ |
| [("batch", "all")] | $\frac{bd_{io}d_h}{n}$ | $d_{io}d_h$ | $\frac{n}{b}$ | $\frac{b}{n}d_{io} + \frac{b}{n}d_h + d_{io}d_h$ |
| [("hidden", "all")] | $\frac{bd_{io}d_h}{n}$ | $d_{io}b$ | $\frac{n}{d_h}$ | $bd_{io} + b\frac{d_h}{n} + d_{io}\frac{d_h}{n}$ |
| [("batch", "rows"), ("hidden", "cols")] | $\frac{bd_{io}d_h}{rc}$ | $d_{io}(\frac{b}{r} + \frac{d_h}{c})$ | $\frac{c}{d_h} + \frac{r}{b}$ | $\frac{b}{r}d_{io} + \frac{b}{r}\frac{d_h}{c} + d_{io}\frac{d_h}{c}$ |
| [("batch", "rows"), ("hidden", "cols"), ("io", "planes")] | $\frac{bd_{io}d_h}{rcp}$ | $\frac{b}{r}\frac{d_{io}}{p} + \frac{b}{r}\frac{d_h}{c} + \frac{d_{io}}{p}\frac{d_h}{c}$ | $\frac{c}{d_h} + \frac{p}{d_{io}} + \frac{r}{b}$ | $\frac{b}{r}\frac{d_{io}}{p} + \frac{b}{r}\frac{d_h}{c} + \frac{d_{io}}{p}\frac{d_h}{c}$ |

Table 1: Computation, communication and memory costs for different layouts of the computation in Algorithm 1. Constant factors and lower-order terms are dropped.

Table 1 shows the computational costs associated with our example computation layouts. The computation time is dominated by that of `einsum` operations. The communication time comes from the *Allreduce* operations, which are necessary whenever the inner dimension of einsum is split. Assuming that the mesh has physical links between all pairs of logically adjacent processors, each *Allreduce* operations can be done in time proportional to the size of one slice divided by the per-link network bandwidth [5].

The network-boundedness of the computation is proportional to the value shown in the table column marked $\frac{communication}{computation}$, with the constant of proportionality depending on the ratio of communication and computation speeds on the given hardware. In the data-parallel layout, the value is $\frac{n}{b}$, the inverse of the per-processor batch size. Performance suffers if the per-processor batch is too small. In the model-parallel layout, the value is $\frac{n}{d_h}$, the inverse of the number of hidden units per processor. Performance suffers if the hidden layer is sliced too finely. For good performance, batch size is irrelevant, but we need the hidden layer to get larger as we increase the number of processors. In the first data-parallel, model-parallel layout, the value is $\frac{c}{d_h} + \frac{r}{b}$. In this layout, we can quadratically increase the number of processors while only linearly increasing the batch size and hidden layer sizes necessary to maintain good efficiency. The final layout lets us cubically increase the number of processors in a 3-dimensional mesh, while only linearly increasing the batch size and the layer sizes.

# 9   Model-Parallel "Transformer"

We implemented a model-parallel layout of the Transformer attention-based sequence-to-sequence model described in [16]. The complete implementation is available in the *tensor2tensor* library on *github*. The layout is given by:

```
mesh_shape = [("all", n)]
computation_layout = [
  ("vocab", "all"), ("d_ff", "all"), ("heads", "all")]
```

That is, the dimensions representing the vocabulary size, the size of the feed-forward hidden layer, and the number of attention heads are each split across all processors. This layout works because every expensive operation in the model has exactly one of these dimensions, and no tensor in the model has more than one. Similarly to the model-parallel layout for our example network (Section 8.2), network-boundedness and memory usage per processor remain constant if we scale all of these dimensions proportionally to the number of processors. We did just this, training transformer models

with ever larger hidden layers and numbers of attention heads on ever larger TPU clusters (we did not increase the vocabulary size). As expected, we saw very similar performance characteristics between the models. This scaling turns out to be highly beneficial to model quality (Section 9.1).

To use even more processors, we combined this model-parallelism with data parallelism, splitting the batch across one dimension of a 2-dimensional TPU mesh and the dimensions described above across the other dimension of the mesh:

```
mesh_shape = [("rows", r), ("cols", c")]
computation_layout = [("batch", "rows"), ("vocab", "cols"),
                      ("d_ff", "cols"), ("heads", "cols")]
```

This layout maintains constant performance if the batch size is scaled proportionally to r and the mentioned model dimensions are scaled proportionally to c. Using this layout, we trained Transformer models with feed-forward hidden dimensions up to 262144 and up to 256 attention heads on 2-dimensional TPUv2 meshes of up to 16x32=512 cores, maintaining computational efficiency of over 50% (6 PFLOP/s out of a maximum 11.5 PFLOP/s) on the largest models.

## 9.1 Experiments and Results

To examine the benefit of scaling the Transformer model in the manner suggested by the previous section, we trained such models on machine translation and language modeling tasks. Results are given in Tables 2 and 3.

For the billion-word language modeling benchmark, we trained the models for 10 epochs. The largest model (4.9B parameters) took 13 hours to train on a 512-core TPUv2 cluster. Batch size for all models was 256 sequences of 256 tokens each (each sequence was the concatenation of multiple training sentences). The batch was split along the mesh dimension of size 16 and the model dimensions were split along the mesh dimension of size 32. Per-word dev-perplexity for the largest model was 24.0, but dropped to 23.5 when the model was evaluated with the logits multiplied by 0.9 (likely due to overfitting). This represents the best published result on this dataset. As expected, perplexity was lower for larger models. We have included random samples from these models in the supplementary materials (Section S.3). On the `languagemodel_wiki_noref_v128k_l1k` dataset from the Tensor2Tensor library[1], consisting of over 5 billion tokens of text from Wikipedia, perplexity continued to improve significantly with a model size of 5 billion parameters.

On the WMT14 En-Fr translation tasks (3), we trained the models for 3 epochs. The largest model (2.9B parameters) was trained for 22 hours on a 128-core TPUv2 cluster. Quality improved with model size, with the largest model achieved BLEU score 43.9 (evaluated using `sacrebleu`), the best published result to date. For the WMT14 En-De dataset, gains from model size were smaller, presumably due to the small size of the training data.

Additional details about the configurations for these experiments are available as part of the `tensor2tensor` library on github.

Table 2: Transformer-Decoder Language Models: $d_{model} = 1024$, $d_k = d_v = 256$

| $d_f f$ | $heads$ | Parameters (Billions) | Billion-Word Benchmark Word-Perplexity | Wikipedia Subword-Perplexity |
|---|---|---|---|---|
| 4096 | 4 | 0.14 | 35.0 | 8.74 |
| 8192 | 8 | 0.22 | 31.7 | 8.03 |
| 16384 | 16 | 0.37 | 28.9 | 7.44 |
| 32768 | 32 | 0.67 | 26.8 | 6.99 |
| 65516 | 64 | 1.28 | 25.1 | 6.55 |
| 131072 | 128 | 2.48 | 24.1 | 6.24 |
| 262144 | 256 | 4.90 | 24.0(**23.5**) | **6.01** |
| Prev Best DNN [15] | | 6.5 | 28.0 | |
| Best DNN Ensemble [13] | | | 26.1 | |
| Best Ensemble (different methods)[13] | | $> 100$ | 23.7 | |

Table 3: Transformer Machine-Translation Results. $d_{model} = 1024$, $d_k = d_v = 128$

| $d_f f$ | $heads$ | $d_k, d_v$ | Parameters (Billions) | WMT14 EN-DE BLEU | WMT14 EN-FR BLEU | |
|---|---|---|---|---|---|---|
| 2048 | 4 | 128 | 0.15 | 25.5 | 41.8 | |
| 4096 | 8 | 128 | 0.24 | 26.5 | 42.5 | |
| 8192 | 16 | 128 | 0.42 | 27.1 | 43.3 | |
| 16384 | 32 | 128 | 0.77 | 27.5 | 43.5 | |
| 32768 | 64 | 128 | 1.48 | 27.5 | 43.8 | |
| 65536 | 128 | 128 | 2.89 | 26.7 | **43.9** | |
| 4096 | 16 | 64 | 0.21 | **28.4** | 41.8 | [16] |

## 10    Related Work

A large part of deep learning computations is a series of matrix multiplications and tensor contractions (*Einsum*s). Distributed matrix multiplication is a well-studied problem in high performance computing. Efficient algorithms partition the computational space, instead of partitioning work by the output matrix/tensor (*owners compute*), to minimize communication. This technique is sometimes called *iteration space tiling* [2], *replication* [6], or *task parallelism* [11]. Mesh-TensorFlow can express a wide range of uniform partitionings of the iteration space and therefore can adopt many best known mappings, e.g., 3D [3, 1] and 2.5D [6] algorithms for square matrices, CARMA [9] for rectangular matrices, 1.5D [14] algorithm for matrices with different sparsities, best tile sizes for direct convolutions [17], etc., although sometimes with higher memory requirements. Furthermore, in most existing work, when multiple multiplications are composed together, the user has to specify the data layout for each matrix separately [22]. Mesh-TensorFlow lets the user name the dimension to split, simplifying the process and allowing for much easier mapping explorations. Feature-wise, Mesh-TensorFlow shares many similarities with the Cyclops Tensor Framework [10], a distributed tensor contraction library originally developed for quantum chemistry applications, which also supports replication and arbitrary mappings.

In the context of deep learning, partitioning the iteration space, e.g., interpolating between data and model parallelism, is relatively new. Gholami et al. [18] analytically showed that using both data and model parallelism at the same time can be more beneficial than using just one of them. Building on top of 1.5D matrix multiplication algorithms, their algorithm can support *replication* and arbitrary processor grid shapes. However, they only explored the parallelization of AlexNet [8] and they have not implemented the algorithm. Jia et al. [20, 19] implemented a framework that uses cost modeling to pick the best parallelization strategy, including how to partition work for each operation. Their *parallelizable dimensions* are defined as the set of all divisible dimensions in the output tensor (*owners compute*), and therefore their mapping can be suboptimal in terms of communication. We expand on this in the supplementary materials (Section S.2).

## 11    Future Work

The Mesh-TensorFlow library is available at `https://github.com/tensorflow/mesh` and is under active development. Some potential areas for development are:

- Automated search for optimal computation layout.
- Implementations of different models and operations. For example, convolutions on spatially-partitioned tensors will require the communication of "halo" regions, as described in [21].
- Implementation of SPMD programming on CPU/GPU clusters.

## 12    Conclusion

In this paper, we introduce the Mesh-TensorFlow language, facilitating a broad class of SPMD distributed tensor computations. Applying Mesh-TensorFlow to the Transformer model, we are able to train models with 5 billion parameters on up to 512-core clusters, establishing new state-of-the-art results for WMT14 En-Fr translation task and the One Billion Word language modeling benchmark.

## Footnotes

[1]No published results exist for this dataset.

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
