[Supplementary Material]

# Supplementary Materials for "Mesh-TensorFlow: Deep Learning for Supercomputers"

**Noam Shazeer, Youlong Cheng, Niki Parmar,**
**Dustin Tran, Ashish Vaswani, Penporn Koanantakool, Peter Hawkins, HyoukJoong Lee,**
**Mingsheng Hong, Cliff Young, Ryan Sepassi, Blake Hechtman**
Google Brain
{noam, ylc, nikip, trandustin, avaswani, penporn, phawkins,
hyouklee, hongm, cliffy, rsepassi, blakehechtman}@google.com

## S.1   Illustrations for the Two Fully-Connected Layers Example

This section provides the illustrations of the four layouts mentioned in the Two Fully-Connected Layers example in Section 8. The overall computation is shown in Figure S.1. We draw a matrix multiplication $C = AB$ by putting $A$ to the left of $C$ and putting $B$ above $C$. For each matrix, we put its name inside, its number of rows on the left or right side, and its number of columns above or below it. We omit the numbers of rows or columns that can be implied from adjacent matrices, i.e., knowing that the multiplication dimensions must match.

Figure S.1: The overall computation of the Two Fully-Connected Layers example. First, $x$ is multiplied with $w$. The temporary result $xw$ is summed with $bias$, applied component-wise $Relu$, and then stored in $h$. Lastly, we multiply $h$ with $v$ to get $y$. The component-wise operations and the intermediate matrix $xw$ are not shown in the figure.

Figure S.2 presents the purely data-parallel layout with $n = 2$ processors. We number the processors 0 and 1, respectively. The ranks of the processors that store each matrix part are written in blue on the matrix part. The matrix names are moved to the bottom-left corners. The whole $w$ and $v$ are labeled with both 0 and 1 because both of them are fully replicated between the two processors. The purely model-parallel layout are drawn similarly in Figure S.3.

Figures S.4 and S.5 show the mixed data-and-model-parallel layout with a 2-by-2 and a 2-by-2-by-2 processor meshes, respectively. We give each processor a serialized rank as shown in the figure, and use the serialized rank to label matrix slices.

Figure S.2: The data-parallel layout for the Two Fully-Connected Layers example, with $n = 2$ processors $\in \{0, 1\}$. Blue numbers on matrices indicate the ranks of the processors the matrix slices reside on. The *batch* dimension is split among all processors. $w$ and $v$ are fully replicated.

Figure S.3: The model-parallel layout for the Two Fully-Connected Layers example, with $n = 2$ processors $\in \{0, 1\}$. Blue numbers on matrices indicate the ranks of the processors the matrix slices reside on. The *hidden* dimension is split among all processors. $x$ and $y$ are fully replicated.

Figure S.4: The mixed data-and-model-parallel layout for the Two Fully-Connected Layers example. There are 4 processors, arranged into a 2-by-2 mesh. Each processor is assigned a serialized rank which is used to label matrix slices that it owns.

Figure S.5: The mixed data-and-model-parallel layout for the Two Fully-Connected Layers example. There are 8 processors, arranged into a 2-by-2-by-2 mesh. Each processor is assigned a serialized rank which is used to label matrix slices that it owns.

## S.2 Per-operation Parallelizations and Communication Costs

Communication is much more expensive than computation and is usually the bottleneck in a parallel program, especially in distributed setting. The section shows how the more common *owner-compute* parallelization strategies can be communication-suboptimal. We start with a simplified overview of the parallelization schemes used in distributed matrix multiplication and tensor contractions (*Einsum*s), focusing on their communication bandwidth costs. (See [1, 2, 4] for rigorous communication lower bounds analyses.) We only discuss distributed matrix multiplication here since the concept is trivially generalizable to its tensor counterpart.

**Iteration Space.** The iteration space is the set of all index tuples required to compute a problem. For example, the matrix multiplication problem $C = AB$ computes $c_{ij} = \sum_k a_{ik} b_{kj}$. Its iteration space consists of all possible tuples $(i, j, k) \in \mathbb{Z}^3, 0 \leq i < u, 0 \leq j < v, 0 \leq k < w$, where $C$ is $u$-by-$v$, $A$ is $u$-by-$w$, and $B$ is $w$-by-$v$, as shown in Figure S.6.

Figure S.6: The iteration space of the matrix multiplication problem $C = AB$. Each voxel $(i, j, k)$ represents the computation $c_{ij} \mathrel{+}= a_{ik} b_{kj}$.

**Parallelization.** Let $n$ be the number of processors. Parallelization corresponds to partitioning the set of voxels into $n$ (not necessarily) equal subsets for each processor to compute. For matrix multiplication, the most widely-used partitionings are grouped into three categories [3]: 1D, 2D, and 3D, based on the number of dimensions of the iteration space that are split. Figure S.7 shows an example for each category. The left image is 1D partitioning ($n = 8$) because only the $j$ axis is split. The middle image splits axes $i$ and $j$ so it is 2D partitioning ($n = 64$). The right image is 3D partitioning ($n = 64$) because all three axes are split.

Figure S.7: Example partitionings of the iteration space for the matrix multiplication problem. The names 1D, 2D, and 3D comes from the number of dimensions that are split.

**Owner computes.** Owner-compute strategies split a matrix (or matrices) equally among processors and each processor is responsible for all computations related to the matrix chunk it *owns*. The 1D and 2D partitionings in Figure S.7 are owner-compute strategies. In 1D case, each processor owns a slice of matrices $B$ and $C$ each, and computes the whole slab requires for its slice of $C$. In 2D case, each processor owns a patch of $C$ and computes a pencil corresponding to it. The 3D partitioning goes beyond owner-compute rule, since no processor is responsible for all computations associated with the data it has. Owner-compute schemes are more common because they are the most intuitive as we often view the output data as the unit of work. We will show why its communication costs are usually suboptimal in the next paragraph.

**Communication.** Here, we focus on the number of elements that have to be transferred by a processor. Let $V$ be the voxel subset assigned to a processor, and $V_A$, $V_B$, and $V_C$ be the projections of $V$ onto the $A$, $B$, and $C$ planes, respectively. The total number of elements a processor has to access to complete its computation is simply $|V_A| + |V_B| + |V_C|$, where $|\cdot|$ denotes set cardinality. Since a processor can only hold a limited amount of data in memory, the rest of the elements must come through communication. The volume of the subset designates the computational workload. As mentioned in the paper, we would like to maximize the computation-to-communication ratio, therefore we want $V$ to have as low surface-to-volume ratio as possible. Assuming $V$ only takes cuboid shapes, then the best shape is a cube.

*Owner-compute* methods fall short when it cannot partition the space into cubes. To illustrate, we compare 2D and 3D partitionings for $p = 64$ processors in Figure S.7. When $u = v = w$, each pencil in the 2D partitioning has a computation-to-communication ratio,

$$r_{\text{2D}} = \frac{2u^3/64}{u^2/64 + 2u^2/8} = 2u/17 \approx 0.12u.$$

Each cube in the 3D partitioning has a higher computation-to-communication ratio,

$$r_{\text{3D}} = \frac{2u^3/64}{3u^2/16} = u/6 \approx 0.17u.$$

**Mesh-TensorFlow.** Mesh-TensorFlow can express more fine-grained parallelism than *owner-compute*, even though all we have to do is just specifying the data layout for each tensor. That is because our layout allows the tensor to be replicated. This, combines with multiple layouts from multiple tensors involved in an operation, can split the iteration space in as many dimensions as necessary (up to the rank of the iteration space).

## S.3   Random Samples from Transformer Language Models Trained on Billion-Word Language Modeling Benchmark

The following samples were randomly generated from the Transformer language models described in the paper. All sentences were seeded with the initial words "According to Ray Kurzweil", and continued randomly by the model. While all the models produce mostly grammatical sentences, the larger models exhibit more world knowledge.

### S.3.1 Model with 0.14B Parameters, PPL=35.0

According to Ray Kurzweil ...

- ... , professor of musical economics at the University of California , Berkeley , the military 's " biggest challenge " might have been growing the region 's growing urban sprawl .

- ... , the chief executive of ITC Emerging Markets , the metals sector should contribute about $ 210 billion by year-end 2008 – a figure that has now risen to $ 3 billion .

- ... , the head of PMI , he has now written off all his employees fairly .

- ... , senior vice president of sales and marketing at Aspen , there is a percentage of the online response for the auto company escalating to 18 customers and increasing to 30 days a year .

- ... , director of the Economic Policy Institute , India ranks 21st out of 221 nations in the current budget deficit , which ranks 13th globally .

- ... , Victor Hesketh 's food agency that produces , sends and distributes the raw beef and pork through the state 's Food and Drug Administration , and they 're also allowed to take the lamb by their own account alongside the chicken and veal .

- ... , the author of " Smoking : The Letters of Hysteria and Reclining the State of South Carolina " ( 2007 ) , 30 percent of liquor 's coconut is sold on the first batch .

- ... , an MIT student who is not involved in anything more than a stock-market move , the latest system of debt and bankruptcy parallels the completely unregulated collection of debt that emerged in the early 1990s , when a financial industry boom brought a financial crisis that spilled almost everywhere from the United States .

### S.3.2 Model with 0.37B Parameters, PPL=28.9

According to Ray Kurzweil ...

- ... , the owner and web guru at Stanford University , even a single person might fall in love with the internet simultaneously .

- ... , PhD , a professor of computer science at Princeton who has staked his reputation on the ability of code technicians to simultaneously amplify a cursor 's legal weight , machines will go digital by using meta-dimensions instead of a brick , and design schemes closer to the core of GPUs .

- ... , chief executive of the company , very few people work through chronology , and most people can 't use tables on their machines .

- ... , we are saving the most jobs in the world .

- ... , creator of the Star Wars creator and creator of the popular game , " Indiana Jones and the Kingdom of the Crystal Skull , " here comes Martin Schafer ( " Spider-Man 3 " ) to describe the businessman as a grand master .

- ... , a technology expert at the Massachusetts Institute of Technology and a host of from academia , Ardipithecus ramidus was frequently incubated with solar-powered cells to frighten away and frighten out the predators , and housed in them so they could damage Lego 's control panels .

- ... , the famously headstrong and egocentric stack-on-stack , a keyboard is more than a part of the laptop 's muscle when it is standing upright rather than crouching , and its outlet at the right end of the screen was probably the key to how the phone turned into its belly .

- ... , founder of the Stanford-funded company , similar " recycled book " concepts are also more of an obsession , as this lets authors track their emotions in print product forms and refuse to look at one all- , -certain identity .

### S.3.3 Model with 1.28B Parameters, PPL=25.1

According to Ray Kurzweil ...

- ... , a transplant surgeon and Harvard astrophysicist , the results are illuminated in Honolulu , Vietnam .
- ... of UNK , a software company in California , computer games are alive and well , with millions of gamers , but most games do not make money .
- ... , the James Watson and James Watson Professor of Physics at MIT , if we all assume that the project will continue to go ahead , we will eventually be in an era when human beings are capable of figuring out just what our genetic make-up is focused on .
- ... , the physicist who has kept many of these principles alive , the world has vanished into " more and more of the static world " – and in many ways we 're losing the earth 's ability to appreciate water .
- ... , creator and the only original idea from the series , the tablet is expected to be a device that combines a USB 2.0 card with an iPad , a laptop and an iPod Touch – with UNK sharp-wired connections to the Internet .
- ...  and a panel of experts testifying in Los Angeles , six years in Congress attempts to improve " brain " of Americans by hitting them with a $ 50 annual fee , then having them likely pay the company $ 3,000 for every additional year in life they are in a job .
- ... , creator of the Review of Medical UNK , the organisation could be the " holy grail " of degenerative brain disease .
- ... , music analyst and co-founder of zero-carbon quantum computing firm Redpoint , if you listen carefully , you 'll hear a far more universal " idea " of how this new car is supposed to be or what it should not be , or how much going to cost .

### S.3.4  Model with 4.9B Parameters, PPL=24.0

According to Ray Kurzweil ...

- ... , chief technology officer for the US Department of Energy , aviation has " potential to be the largest and fastest growing source of consumer and commercial emissions . "
- ... , the futurist son of the futurist who wrote The Singularity is Near , the " early days " of Google are not the right time to push forward with the next great leap .
- ... , creator of the first modern computer , the sexy cyborg was the brainchild of an MIT professor , Thomas Harris , and a former banking entrepreneur , Henry Lee , who was looking for an UNK , a light that could be recovered and used to light up the Internet .
- ...  , the inventor of the modern personal computer , the shrinking human brain could eventually replace the Internet as a tool of human intelligence and imagination .
- ... , the expert and co-author of " The Singularity is Near : Comprehending the Technological Future of Engineering , " people are looking for ways to protect and make their lives better .
- ... , creator of the Gaia hypothesis , earlier computer systems should become not just more efficient , but more-efficient , increasing their efficiency by reducing human errors ( the unexpected , but often the regrettable ) and even the number of errors .
- ... , who will make an appearance at this year 's Consumer Electronics Show in Las Vegas next week , these mobile gadgets will be able to " talk " to each other .
- ... , the futurist turned futurist , the onset of Alzheimer 's coincided precisely with the rate of unemployment in America .