[Reviews · NeurIPS 2018]

Reviewer 1



This paper has an interesting explanation and analysis of a tensor-oriented implementation of model parallel training of huge models. Personally I would have preferred an explanation that was a bit more "code-like" and looked less like equations, but I suppose a certain 'look' is expected for NIPS papers. The state-of-the-art results for language modeling are also very interesting and probably warrant inclusion in NIPS. (I don't have anything particular to add after seeing the reviewer feedback, since they were addressing points from other reviewers. Leaving my review as-is).

Reviewer 2



Overall. The paper is about introducing a new language of abstraction for distributed tensor operations, with a focus on shallow, feed-forward neural networks. Specifically, the paper describes the concept of laid-out tensors, which are "sub-tensors" that can either stand for weights or data and be distributed among different processors to allow for data- or model-parallelism. A new symbolic language for computations with laid-out tensors is described and is shown to be convenient for describing a big variety of possible model constructs, involving model and/or data parallelisms. A theoretical performance analysis is provided, explaining how to avoid "wasted" resources when defining models and data in parallel. Experiments are performed on TPUs obtaining improved results on models that run on 256 cores on language tasks. Details. The main question about the paper is whether it is a good fit content-wise to NIPS. While distributed machine learning is clearly relevant, the paper focuses mostly on describing a new abstraction layout for making the computations and the models more scalable. Perhaps, as a work it would be more relevant to a conference on distributed systems. From a technical point of view, what seems to be unclear is whether the proposed language is also applicable to more complex architectures, like multi-layered neural networks, recurrent neural networks, or DAG neural networks. Will the proposed language be sufficient when there must be increased communication between interweaved layers, or increased communication between layers over time? For instance, what if there are states that need to be stored and communicated between nodes. What is further unclear is what is the effect of the mesh shape on the final computations. Would the theoretical costs provided in Table 1 change for different mesh shapes? Or is this already determined by the respective layout (leftmost column in Table 1). Last, from the experimental point of view it would be interesting to have some deeper analysis. First of all, are there any benefits regarding the training time, or the only improvement is on the growing of the models? Also, are the theoretical costs from Table 1 also verified experimentally? Is it possible to provide a similar table or figure that compared the theoretical costs with the empirical ones, to validate the proposed language abstraction? Regarding the writing, the paper could improve its clarity at points. - It would be useful to summarize all notations in a table. - Providing a schematic of the model that is used to train for the experiments would help reading. - Giving a better explanation on pleasantness, as it seems to be an important concept later on. Perhaps a graphical illustration can help the reading. - Regarding Table 1, it would be useful to explain how are these costs exactly derived?

Reviewer 3



Given the author's additional information about their plans to open-source the language described in this paper, I've upgrade my review. I'm still concerned that the original paper didn't mention anything about this intent to open-source; but I'm willing to give them the benefit of the doubt because if this is open-sourced, it could be very helpful to the community. Here are some minor typos I found: Capitalize “equation”, “table”, “section” as needed throughout the paper. Line 112: “pleasant” to “pleasantly” (and throughout the paper where needed) 219: “tpu” to “TPU” 222: “(3)” to “Table 3” ORIGINAL REVIEW This paper introduces a convenient notation for compactly describing mappings from a tensor representation of a learning problem’s data and model parameters to an n-dimensional mesh of compute nodes. The notation is flexible enough to capture both data-parallel and model-parallel mappings, as well as mixtures of the two. The paper uses these mappings to derive first order approximations to the computation and communication times required to perform a single backpropagation update (assuming an optimized communication library like MPI). The paper then uses these estimates to develop a strategy for simultaneously growing the model size and the number of compute nodes used such that communication does not become a major performance bottleneck. The paper then uses this strategy to explore the parallel performance of the Transformer sequence transduction model. The strategy allowed the authors to train a 5B parameter system on 256 nodes and get state-of-the-art performance on the WMT14 En-Fr data set. When I first read this paper, I thought that the authors meant that they had created a new programming language for easily and efficiently mapping machine learning problems to parallel computer nodes. I was hoping that this “language” would be made available to the reader and that it would include some automatic optimization of the mappings; but as I read more of the paper, I realized that the paper only presented a notation for representing the mapping. If a new programming language (or tool) has been created, the authors should make that more clear and maybe give pointers to it. If it is only a notational convenience, the authors should make that more clear too. If the contribution of Section 4 is only to introduce a notational convenience, the authors should explain the significance more clearly and/or concretely. For example, the notation in Algorithm 1 is simply Einstein Summation Notation in another form. It doesn’t add significantly to the reader’s understanding of the problem. Furthermore, the observations made in Table 1 are fairly well-known from a parallel computing point of view - most practitioners realize this computation/communication trade off. And for those NIPS readers who don’t, the paper has no explanation for where these numbers come from. The paper could be improved if Section 4 was much shorter and the authors spent time explaining how they calculated the terms in Table 1. It seems that the major result of this paper is a better BLEU score from the ability to run a bigger model. I’m not sure that is significant enough. The paper would benefit from clarification in several places. For example, “b”, “d_x” and “d_y” are never explicitly defined. Equation (3a) is described and then never used again in the paper. The paper’s “{b-->0}” notation is never explicitly defined. Presumable “0” means dimension zero of the compute mesh; but it would be better it the authors specifically said it so that the reader doesn’t have to guess. Likewise, “F” and “D(X,M)” are never defined. (I know “F” is an arbitrary function, but the paper should say that instead of making the reader guess.) Also, for example, when two model dimensions map to a single mesh dimension, does that mean the mesh dimension is split 50/50? It could be split based on some communication minimization, or other, algorithm. The reader has to guess. The text mentions four layouts but only shows three in Table 1. In Table 3, why are two of the BLEU scores missing? Was it that they couldn’t run? Ran too slow? Wasn’t enough time? That should be explain. In Section 5, the reader would benefit from some pictures showing, for example, the matrix multiplication, etc., to help them “see” the mapping. “SOTA” should be replaced with “state-of-the-art” throughout the paper.